# Ultrafast all-optical second harmonic wavefront shaping

Artem Sinelnik [1,2,8], Shiu Hei Lam[2,8], Filippo Coviello [1,2,3,8], Sebastian Klimmer [1], Giuseppe Della Valle [3,4], Duk-Yong Choi [5], Thomas Pertsch [2,6,7], Giancarlo Soavi [1] & Isabelle Staude [1,2,7] ✉

Optical communication can be revolutionized by encoding data into the orbital angular momentum of light beams. However, state-of-the-art approaches for dynamic control of complex optical wavefronts are mainly based on liquid crystal spatial light modulators or miniaturized mirrors, which suffer from intrinsically slow (μs-ms) response times. Here, we experimentally realize a hybrid meta-optical system that enables complex control of the wavefront of light with pulse-duration limited dynamics. Specifically, by combining ultrafast polarization switching in a WSe$_2$ monolayer with a dielectric metasurface, we demonstrate second harmonic beam deflection and structuring of orbital angular momentum on the femtosecond timescale. Our results pave the way to robust encoding of information for free space optical links, while reaching response times compatible with real-world telecom applications.

Three decades ago, Allen et al. demonstrated that, in addition to linear momentum and spin angular momentum, photons propagating in paraxial vortex beams possess orbital angular momentum (OAM)[1,2], or topological charge, which takes values that are integer multiples of the reduced Planck constant. The OAM is associated with the global structure of light and the topological charges correspond to the number of intertwined helical surfaces in the wavefront of a light field[3–5]. Importantly, OAM light beams with different integer topological charge represent orthogonal states, even for finite apertures and regardless of the aperture size[6]. As a consequence, they are exceptionally robust against perturbations, making them a viable platform for the encoding of digital information of both classical and quantum nature[7–10], and an interesting candidate to replace commonly used amplitude or combined amplitude/phase encoding schemes. In particular, OAM beams hold great potential for free space optical communications, where atmospheric perturbations pose an omnipresent challenge[7]. Moreover, it is impossible to recover the topological charge from the light scattered by the atmosphere, because the time-dependent scattering processes randomize the phase structure of the beam[11]. Consequently, OAM encoded information is resistant to eavesdropping, adding the paramount feature of security to its potential for free-space optical communication.

While being highly promising, the main missing piece to the spread of free-space optical communication based on encoding of digital information into OAM is the lack of suitable methods for ultrafast wavefront shaping, which is required for the dynamic generation of topological charges, as opposed to the simpler modulation of intensity or temporal phase. As a consequence, the speed at which information can be encoded via the OAM is still limited. To date, the most common approach was based on liquid crystal (LC) spatial light modulators (SLMs)[12] with response times of tens of ms, restricting data rates to the kbit/s range[13]. To reduce the response times down to μs, technologies such as ferroelectric LC-SLMs[14], digital micromirror devices (DMDs)[15] and photothermal SLMs[16] have been suggested. DMDs in particular have already been used for OAM encoding with a record high modulation rate of 17.8 kHz[17], which is however still far from the demands of realistic telecom applications. For this reason, light structuring has been

[1]Institute of Solid State Physics, Friedrich Schiller University Jena, Jena, Germany. [2]Abbe Center of Photonics, Institute of Applied Physics, Friedrich Schiller University Jena, Jena, Germany. [3]Dipartimento di Fisica, Politecnico di Milano, Piazza Leonardo da Vinci, 32, Milano, Italy. [4]Istituto di Fotonica e Nanotecnologie, Consiglio Nazionale delle Ricerche, Piazza Leonardo da Vinci, 32, Milano, Italy. [5]Laser Physics Centre, Research School of Physics, Australian National University, Canberra, ACT, Australia. [6]Fraunhofer Institute for Applied Optics and Precision Engineering, Jena, Germany. [7]Max Planck School of Photonics, Jena, Germany. [8]These authors contributed equally: Artem Sinelnik, Shiu Hei Lam, Filippo Coviello. ✉e-mail: isabelle.staude@uni-jena.de

exploited for multiplexing, but not for the encoding of optical bits in telecom applications[18].

As such, the possibility to switch vortex beams at GHz rate remains a major task for the success of OAM-based communication and information technology. More broadly, the capability to control the wavefront of light at ultrafast time scales would also enable disruptive developments in other technological areas, such as beam steering, quantum optics, structured illumination microscopy, or LIDAR applications. In this respect, exciting new opportunities are offered by all-optically tunable metasurfaces[19–25], which exploit the near-field enhancement of their resonant building blocks to manipulate different ultrafast nonlinear and optoelectronic effects. However, designing spatially variant structures for ultrafast complex wavefront control is difficult and thus, most existing works are limited to intensity modulation effects in periodic, homogeneous structures. As notable exceptions, Vabishchevich et al. demonstrated picosecond all-optical diffraction switching[26]; Iyer et al. achieved sub-picosecond steering of ultrafast incoherent emission[27]; Shaltout et al. combined a passive metasurface with a frequency-comb source to realize laser beam steering with a continuously changing steering angle[28]; and Di Francescantonio et al. realized all-optical routing of telecom photons upconverted to the visible range[29]. However, ultrafast all-optical wavefront control with high spatial complexity as e.g., required for vortex beam switching, has not been realized so far.

A new avenue for ultrafast all-optical switching is offered by the intriguing nonlinear optical properties of monolayers of transition metal dichalcogenides (TMDs)[30]. Their crystal symmetry, combined with their atomically thin nature which relaxes phase matching constraints, allows for ultrafast all-optical modulation of the second harmonic (SH) field[31], at switching speed only limited by the pulse duration. Considering the $D_{3h}$ symmetry of TMD monolayers, the emitted SH intensity in the basis of the armchair (AC)/zigzag (ZZ) axes can be written as a function of the incident electric fields along these crystallographic directions as[31]:

$$\begin{pmatrix} I_{AC}(2\omega) \\ I_{ZZ}(2\omega) \end{pmatrix} \propto \begin{pmatrix} \left| E_{AC}^2 - E_{ZZ}^2 \right|^2 \\ 2\left| E_{AC} E_{ZZ} \right|^2 \end{pmatrix} \quad (1)$$

On the basis of Eq. (1), the polarization of the incident fundamental pulses can be used to manipulate the polarization of the SH signal: if the time delay $\Delta t$ between the two orthogonally polarized excitation pulses exceeds their pulse duration $\tau$, the SH signal will be polarized along the AC direction $(I_{AC}(2\omega) \neq 0, I_{ZZ}(2\omega) = 0)$. On the other hand, the maximum SH emission along the ZZ direction is achieved for a delay $\Delta t = 0$, where the two fundamental pulses overlap in the time domain, leading to vanishing SH emission along the AC direction $(I_{AC}(2\omega) = 0, I_{ZZ}(2\omega) \neq 0)$. However, despite its far-reaching potential, the mechanism has up to now only been employed for polarization and amplitude modulation[31].

We highlight that our approach for ultrafast polarization switching[31], which provides one building block of the final device presented here, only works in absence of phase matching constraints. Thus, while the atomically thin TMD could be replaced by ultra-thin films (e.g., few layer 3R-TMDs[32]), our approach for ultrafast SH polarization switching is prohibited in bulk materials. Furthermore, compared to metasurfaces based on thin films of III–V semiconductors[29] or other conventional nonlinear materials, 2D materials offer distinct additional perspectives and advantages: ease of integration on-chip[33], fiber[34], and nanostructured photonic architectures[35], excitonic response and tunability at room temperature[36], selective interaction with light beams carrying spin[37] and/or orbital[38] angular momentum.

Here, we demonstrate ultrafast all-optical switching of complex wavefronts using a cascaded meta-optical system consisting of a monolayer TMD, a quarter-wave plate, and a silicon metasurface.

Specifically, we perform spatially resolved pump-probe experiments to demonstrate the possibility of realizing second harmonic beam deflection, Gaussian-to-vortex beam switching and topological charge switching down to femtosecond time scales. Importantly, given the capability of polarization-selective metasurfaces to imprint also more complex spatial phase profiles[39], our approach has the potential to enable switching between any two wavefronts having nearly arbitrary intensity profiles and pulse-duration limited dynamics. Our results can thus enable new technological developments in the fields of high-speed communications, remote sensing, ultrafast optics, and holographic techniques.

## Results and discussion
### Concept of ultrafast all-optical wavefront switching
A sketch illustrating the proposed ultrafast all-optical wavefront switching scheme is presented in Fig. 1. We use a WSe$_2$ monolayer to generate SH, where the linear SH polarization is controlled by the delay between two synchronized femtosecond laser beams with orthogonal linear polarization oriented along the AC and ZZ directions of the crystal. WSe$_2$ was selected due to the low energy of its excitonic resonances as compared to other materials of the 2D TMD family, allowing for resonantly enhanced SH emission near 750 nm and thus facilitating metasurface fabrication. For delay values $\Delta t$ exceeding the pulse duration $\tau$, the SH is polarized along the AC direction, while for simultaneous arrival of the two pulses ($\Delta t = 0$), the SH is polarized along the ZZ direction[31]. Using a quarter-wave plate, the linear AC (ZZ) polarization of the SH beam is converted into left (right)-handed circular polarization. Finally, the circularly polarized SH light is transmitted through a polarization-sensitive wavefront-shaping metasurface and the output is recorded using a CCD camera. Specifically, we target three different effects of ultrafast wavefront shaping, namely beam deflection, Gaussian-to-vortex beam switching, and topological charge switching.

In order to experimentally implement the described scheme, in a first step a WSe$_2$ monolayer was produced by mechanical exfoliation[40]. A corresponding optical microscope image is shown in Fig. 2a. To confirm that the exfoliated crystal is indeed a monolayer as well as the absence of notable strain in the crystal, we further conducted photoluminescence spectroscopy and pump-polarization-dependent SHG measurements. Figure 2b, c summarize the polarization-resolved SH spectra of the monolayer for two different delay times $\Delta t$. As expected, for $\Delta t > \tau$ (Fig. 2b), the generated SH is linearly polarized along the AC direction, while for $\Delta t \approx 0$ (Fig. 2c), it is linearly polarized along the ZZ direction. Additional optical characterization of the WSe$_2$ monolayer can be found in Supplementary Fig. 5. To shape the wavefront in a polarization-sensitive fashion, metasurfaces composed of hydrogenated amorphous silicon nanoresonators were designed based on the principle of Pancharatnam-Berry phase[41]. For the metasurface supporting Gaussian-to-vortex beam switching, a spatially variant propagation phase[39] was additionally implemented in the design, as it requires an additional degree of freedom. This is the case since the local phases for the two different polarizations are not connected via a simple sign flip as for the other considered phase profiles. Conveniently, the PB phase for a given unit cell design changes the sign but preserves the amplitude for orthogonal circular polarizations, such that a simple reversal of a phase gradient, as required for the metasurfaces implementing beam deflection and topological charge switching (see below for details), is by default implemented. More details can be found in the section "Design principle of the metasurfaces" in the Supplementary Information. All three types of metasurfaces were fabricated on a glass substrate using electron-beam lithography and inductively coupled plasma etching using an established process[42]. For each of the three targeted wavefront shaping effects, a separate metasurface with its own unique geometry and corresponding spatially variant phase profiles

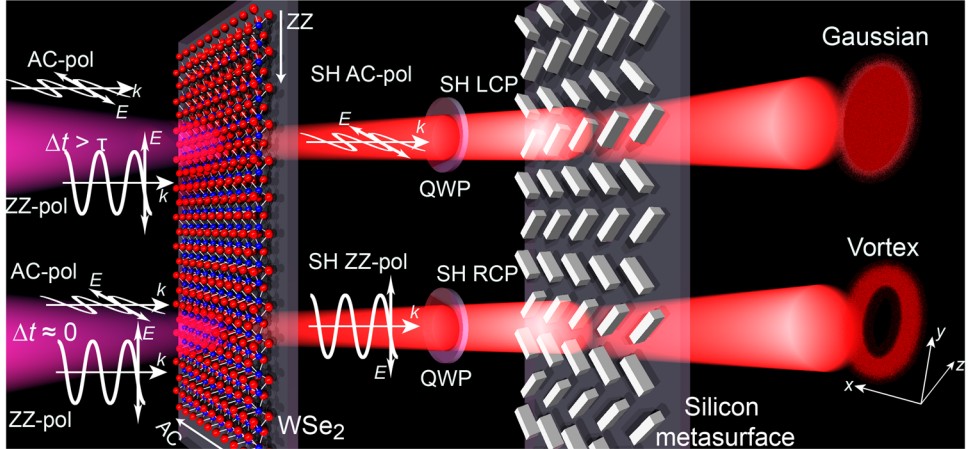

**Fig. 1 | Schematic representation of the operating principle of the cascaded TMD-metasurface structure for ultrafast wavefront shaping.** In the first step, the polarization axes of two orthogonally linear polarized pulse replicas with the same pulse duration $\tau$ are aligned along the main crystal axes (armchair (AC) and zig-zag (ZZ) directions, respectively) of the WSe$_2$ monolayer. As a result of the D$_{3h}$ symmetry of the crystal lattice and the associated contributions of the nonlinear susceptibility, depending on the time offset ($\Delta t$) of the two pulse replicas, the signal of the generated second harmonic (SH) in the WSe$_2$ monolayer is emitted either along the AC ($\Delta t > \tau$) or the ZZ direction ($\Delta t = 0$). In the second step of the cascaded structure, a quarter-wave plate (QWP), whose fast axis is oriented at a 45° angle with respect to both, AC and ZZ direction, leads to a left (right) handed circular polarization of the SH for an emission along AC (ZZ), depending on the temporal delay in the aforementioned SHG process. Finally, the designed silicon metasurface manipulates the SH wavefront depending on the helicity of the incident radiation, which, in the displayed example, leads either to a Gaussian or a vortex beam shape. $E$ and $k$ denote the electric field and the wave vector, respectively. "pol" is the abbreviation of "polarization".

was fabricated. Figure 2d–f shows the target phase profiles for both circular polarizations, as well as the light microscope images and scanning electron micrographs of the fabricated metasurfaces. The nanoresonators of all metasurfaces have an elliptical cross section with a height of 475 nm.

## Numerical simulations

Next, we numerically simulated the far-field distributions in Fourier space of a right-handed (RCP) and left-handed (LCP) circularly polarized Gaussian beam transmitted through the polarization-dependent metasurfaces using the commercial software package FDTD Lumerical (Fig. 3a–f). The metasurface designed for ultrafast beam deflection implements the spatial phase profile of a blazed grating with the sign of the blaze reversing with handedness of the illumination polarization. Accordingly, the incident beam is dominantly channeled into the first diffraction orders, where the sign of the diffraction order changes with handedness (Fig. 3a, d). For the metasurface sculpted for Gaussian-to-vortex beam switching, the spatial phase distribution for RCP light is flat. For LCP light, an azimuthal phase gradient ranging from 0 to 2π is implemented, resulting in the formation of a vortex beam with topological charge $l = 1$. As expected, the intensity distribution behind the metasurface changes from a Gaussian distribution to a donut shape with a dark zone at the center caused by the topological singularity at the beam axis[1] (Fig. 3b, e). Finally, the third metasurface is designed to implement an azimuthal phase gradient ranging from 0 to 2π for both incident polarizations, where the direction of the gradient reverses with handedness[43], corresponding to the formation of vortex beams with $l \pm 1$, respectively. Note that for topological charge switching simple observation of the far-field distributions is not sufficient, since vortex beams with $l \pm 1$ have identical far-field intensity patterns. Thus, to evaluate the sign of $l$ of the output beam, we employ an astigmatic transformation, through which the intensity pattern of the vortex beams acquires the form of dark stripes, whose orientation changes with the sign of $l$ (Fig. 3c, f)[44,45] (see Supplementary Information section on "Theoretical calculation of vortex beam propagation through a cylindrical lens"). Experimentally, this transformation can be implemented by observing the intensity distribution in the focal plane of a cylindrical lens[44].

## Experimental demonstration

For an experimental demonstration of the various ultrafast wavefront shaping effects, we constructed a dedicated pump-probe setup featuring a Fourier microscope to observe the ultrafast change in the wavefront. Supplementary Fig. 3 shows a sketch of the pump-probe setup used to observe the ultrafast change in the wavefront. As light source, we used the signal beam of an optical parametric oscillator (OPO) laser system (Mira OPO, APE, Germany) pumped by a femtosecond Ti:Sa laser (Coherent Chameleon Ultra II). The output wavelength was set to 1500 nm, the pulse repetition rate was 80 MHz, the pulse duration was 200 fs, and the power was 10 mW. We used a commercial common-path birefringent interferometer (GEMINI, NIR-EOS) to create two beams with orthogonal polarization and to control the time delay between them with attosecond precision and subwavelength interferometric stability. The superimposed beams were focused onto the WSe$_2$ monolayer using a Mitutoyo NIR M Plan 10x NA = 0.26 objective, and the SH signal was collected using an identical objective. A short pass filter was used to filter out the beam at the fundamental harmonic frequency. Next, to convert the linear polarization of the SH beam (AC or ZZ) into circular polarization (LCP or RCP, respectively), a quarter-wave plate was employed. It should be noted that the orientation of the quarter-wave plate was fixed during the experiments, the polarization was changed only by the ultrafast nonlinear polarization switching effect in the WSe$_2$ monolayer as a function of the pulse delay. The circularly polarized SH beam was focused onto the metasurface using a Mitutoyo NIR M Plan 10x NA = 0.26 objective, the transmitted signal was collected with a 20x NA = 0.4 objective from the same manufacturer. Finally, a system of lenses was used to image the back focal plane of the collection objective onto an sCMOS camera (prime 95B, Teledyne, USA). For the case of topological charge switching, one of the lenses was replaced by a cylindrical lens. This lens introduces a controlled astigmatism, leading to different intensity distributions on the camera, depending on the sign of $l$. Figure 3g–l shows the recorded intensity distributions in the far field for different delay values $\Delta t$ corresponding to subsequent ($\Delta t > \tau$, left) and simultaneous ($\Delta t \approx 0$, right) arrival of the two pulses. For all three studied metasurfaces, the targeted ultrafast switching effects can be observed. As a manifestation of ultrafast beam deflection, Fig. 3g, j

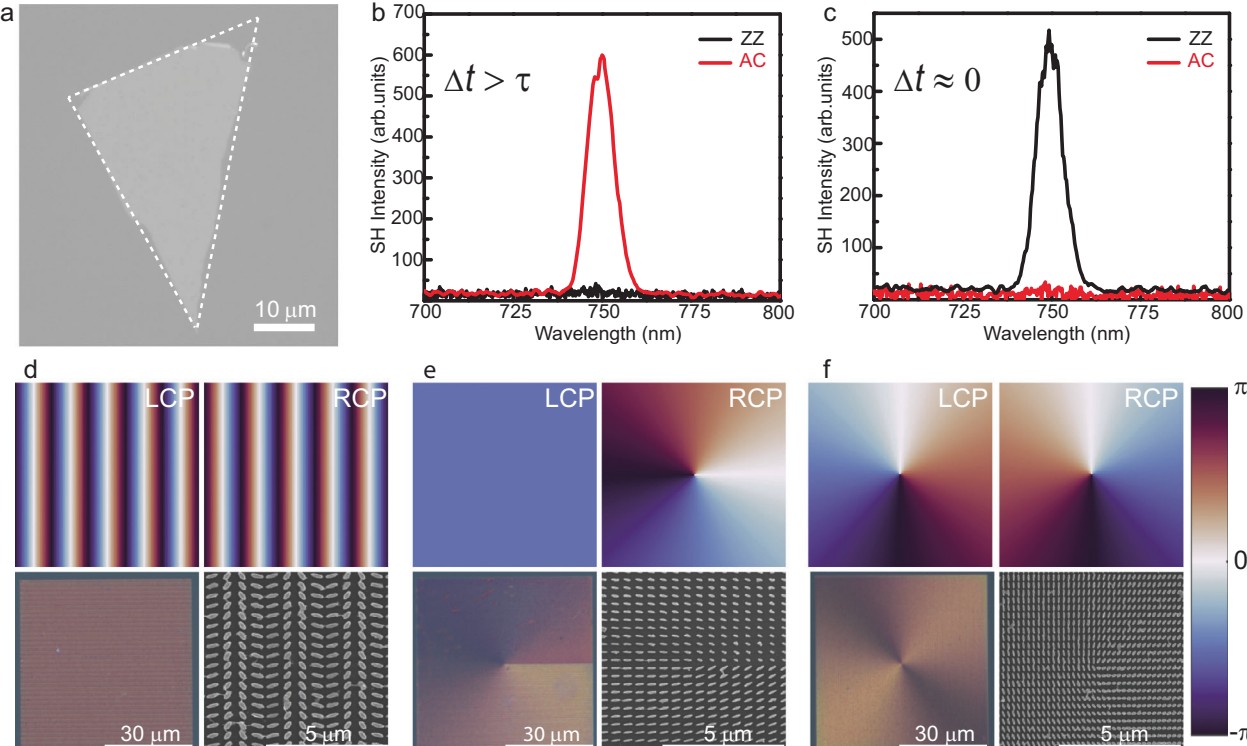

**Fig. 2 | Components of the fabricated hybrid meta-optical system. a** Optical microscope image of the employed WSe$_2$ monolayer. The white dotted line highlights the boundaries. **b**, **c** SH spectra of the WSe$_2$ monolayer for $\Delta t > \tau$ (**b**) and $\Delta t \approx 0$ (**c**). **d**–**f** The top row shows the phase distribution for different circular polarizations of the incident field. The bottom row shows optical images of the metasurfaces (left) and SEM images of the central areas of the structures (right) for **d** beam deflection, **e** Gaussian-to-vortex beam switching, and **f** topological charge switching, respectively.

shows the ultrafast change in the intensity channeled into the first diffraction orders of different sign. To determine the diffraction efficiency, we calculated the ratio of the SHG intensity channeled into the relevant first diffraction order ($+1$ ($-1$) for $\Delta t \sim 0$ ($\Delta t > 0$), respectively) and the total transmitted SHG intensity, yielding $0.55 \pm 0.05$ ($0.57 \pm 0.05$), correspondingly. Moreover, taking the ratio between the intensities in the $+1^{st}$ ($-1^{st}$) diffraction orders for both delay settings allows us to quantify the switching contrast as $0.95 \pm 0.05$ ($0.95 \pm 0.05$). Figure 3h, k shows the change from a Gaussian to a donut-shaped intensity distribution as the temporal pulse overlap is established, thereby providing clear evidence of ultrafast Gaussian-to-vortex beam switching. Finally, Fig. 3i, l shows intensity distributions characteristic for switching of the topological charge from $l = -1$ to $l = +1$ by changing the pulse delay[44,45]. Deviations of the measured intensity distributions from the theoretical expectations can be explained by fabrication imperfections of the metasurface sample. For example, for beam deflection, a larger amount of light is also channeled into the zeroth diffraction order for both displayed settings of the delay stage.

Finally, in order to directly monitor the ultrafast dynamics of the observed complex wavefront shaping effects, we recorded the far-field intensity as a function of the delay time $\Delta t$ for the case of Gaussian-to-vortex beam shaping. To ease the representation, instead of the full two-dimensional image, a cross section through the center of the beam is displayed. Note that for overlapping pulses, the polarization of the GEMINI output changes systematically between linear, elliptical, and circular polarization as the delay time is changed by less than an optical cycle. The full interferometric trace is shown in Fig. 4a, additional information on the polarization of the output provided in Supplementary Fig. 4. For further discussion, from the full interferometric trace we extracted only those frames which correspond to a linear output polarization (Fig. 4b). Clearly, if the delay is less than the pulse

duration (i.e., $\Delta t < -100$ fs), we observe the typical cross section of a vortex beam, namely two maxima and a central minimum of intensity. In contrast, when the delay exceeds the pulse duration, the cross section is characterized by a single intensity maximum typical of a Gaussian beam.

In conclusion, we experimentally demonstrated femtosecond all-optical wavefront control using polarization-dependent wavefront-shaping metasurfaces. This was accomplished by combining all-optical second-harmonic polarization switching from a monolayer TMD with a polarization-sensitive wavefront shaping metasurface. We demonstrated three different wavefront shaping effects, namely beam deflection, Gaussian-to-vortex beam switching and topological charge switching, with response times in the femtosecond range. Our approach allows for switching between any two arbitrary intensity profiles with pulse-duration limited dynamics and thus opens a pathway towards ultrafast OAM encoding of information for free space optical communication links. Specifically, we envision that in a setting, where two stabilized and orthogonally polarized laser beams arrive simultaneously at the meta-optical system, the latter can transform any amplitude information encoded in one of the beams into a complex-wavefront encoded information without any delayed response down to fs time scales. Furthermore, and beyond the advantage in terms of high operation speed, we highlight that OAM states of light, compared to amplitude modulation, are exceptionally good performers for the encoding of digital information, being highly resistant to eavesdropping and offering a level of security that does not depend on mathematical or quantum-mechanical encryption methods[11]. More generally, the demonstrated speed-up in the synthesis of complex arbitrary wavefronts by more than 6 orders of magnitude offers wide-reaching opportunities for both fundamental science and applications in high-speed communications, remote sensing, ultrafast optics and holographic techniques. Importantly, since the underlying SHG

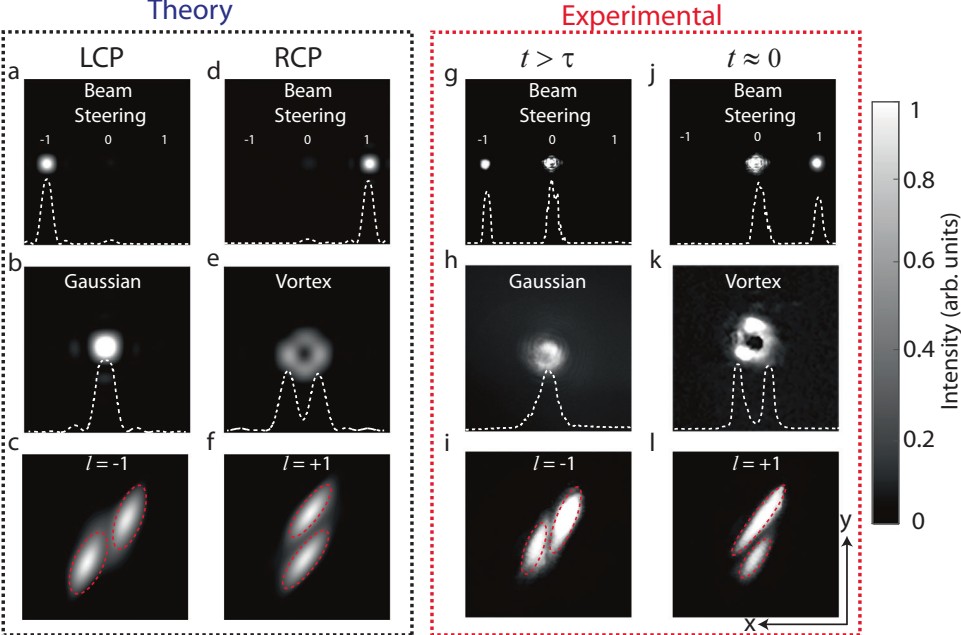

**Fig. 3 | Theoretical (left) and experimental (right) far-field distributions.**
**a**–**c**, **g**–**i** LCP and **d**–**f**, **j**–**l** RCP light transmitted through the polarization-dependent metasurfaces in Fourier space. The first row shows results for beam deflection, the middle row for Gaussian-to-vortex beam switching, and the bottom row for topological charge switching. For (**c**), (**f**), (**i**), (**l**) propagation of the vortex beam through a cylindrical lens was additionally considered.

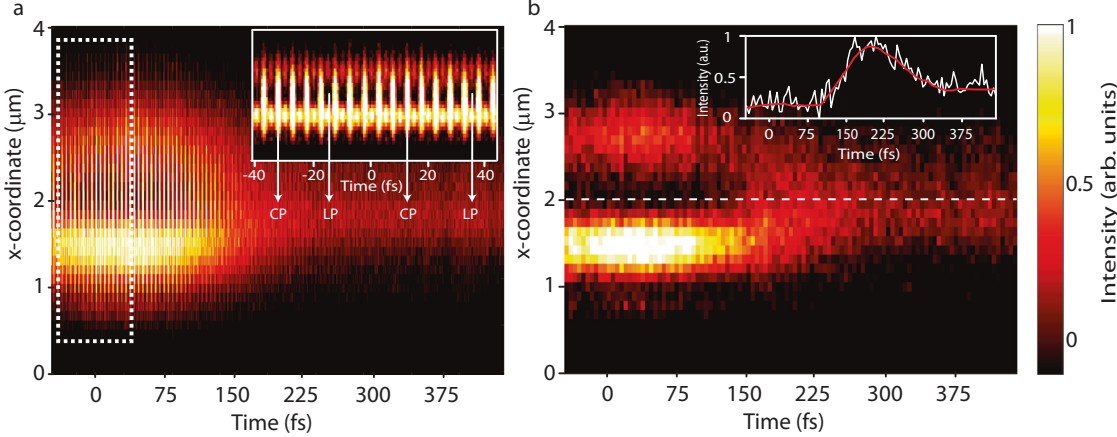

**Fig. 4 | Ultrafast wavefront shaping dynamics. a** Cross section of the far-field intensity as a function of the delay time $\Delta t$ for the case of Gaussian-to-vortex beam switching (full interferometric trace). The inset shows a fragment of the interferometric fringes obtained from the region marked by the white dashed lines. **b** Reduced interferometric trace for frames corresponding to a linear output polarization only. The inset shows the intensity in the center of the map (dashed white line) versus time. The red curve represents a guide to the eye obtained by smoothing the experimental data using a Savitzky–Golay filter.

characteristics are a consequence of the crystal structure of the $WSe_2$ monolayer and thus broadband, the demonstrated scheme can be scaled to other wavelengths—albeit with varying SH generation efficiency—by simple adjustment of the metasurface geometry. Also, an increase of SH output power could be achieved within the limits dictated by phase-matching considerations by replacing the $WSe_2$ monolayer by a bulk $3R$-$WSe_2$ (rhombohedral) crystal.

## Methods
### Numerical simulations
The simulations of the far field of the metasurfaces were performed using the commercial software package FDTD Lumerical. Each metasurface consists of 20 by 20 nanoposts implementing the required phase profile and resting on a glass substrate. The truncation of the metasurface allows fast computation, while retaining a reasonable

representation of the physical system. In the simulations, a circularly polarized pulse with plane wave profile excites the metasurface in a domain enclosed by perfectly matched layer (PML) boundaries. The output field is recorded and Fourier-transformed to provide the spectral static field distribution. The far-field patterns of the metasurfaces were projected from the near field at the plane immediately after the nanoposts.

The calculation of the far field of vortex beam propagation through a cylindrical lens is performed with diffraction theory (Details in the corresponding section of Supplementary).

### Fabrication of the metasurfaces
The proposed metasurfaces were fabricated through a meticulous process on a 1 mm-thick fused silica substrate. Before fabrication, the substrate was cleaned using a combination of acetone, isopropyl

alcohol, and deionized water in an ultrasonic bath to enhance its adhesion to the a-Si:H film. Following this, a 475-nm-thick layer of a-Si:H was deposited onto the substrate using $SiH_4$ (25 sccm) gas with helium (475 sccm) dilution in a plasma-enhanced chemical vapor deposition system (Plasmalab 100 from Oxford). The substrate temperature, operating pressure, radio-frequency power at 13.56 MHz were 200 °C, 1900 mTorr, and 40 W, respectively. To define the desired features, a positive electron beam resist, ZEP520A from Zeon Chemicals, was spin-coated onto the substrate. In addition, Espacer (300Z from Showa Denko) was applied to prevent charging during the electron-beam exposure. The nanostructures were written into the resist by electron beam lithography (Raith150), followed by the development in ZED-N50. A 45-nm-thick aluminum film was deposited onto the substrate through electron-beam evaporation (Temescal BJD-2000) and then patterned by removing the resist with a solvent (ZDMAC from Zeon Co.). These aluminum patterns served as a robust etch mask during a-Si:H plasma etching. The etching process was carried out using a fluorine-based inductively coupled-plasma reactive ion etching system (Oxford Plasmalab System 100). The etching condition was optimized by controlling the mixture of $CHF_3$ and $SF_6$ gases, pressure, and bias/induction powers which resulted in the formation of nanopillars with a high aspect ratio and vertical sidewalls. Subsequently, aluminum wet etching solution was used to remove any residual aluminum from the patterned nanopillars.

## Fabrication of the WSe$_2$ monolayer

Tungsten diselenide ($WSe_2$) monolayers were mechanically exfoliated from a commercially available bulk crystal (HQGraphene). The monolayers were subsequently transferred onto a fused silica substrate by a deterministic dry-transfer technique using a poly-dimethylsiloxane (PDMS, GelPak PF-40-X4) stamp. The monolayers were pre-characterized in a custom-built optical microscope by photoluminescence spectroscopy as well as optical contrast measurements.

## Data availability

Data that support the plots within this paper and other findings of this study are available from the corresponding author upon reasonable request. The fabrication layouts (https://doi.org/10.6084/m9.figshare.25112702.v1) of the three metasurfaces and the simulation models (https://doi.org/10.6084/m9.figshare.25112717.v1) used in this study are available on Figshare.

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

## Acknowledgements

We acknowledge Marijn Rikers for help with SEM images. A.D.S. acknowledges Dr. Ivan Shishkin for the discussion on optical measurements. This project has received funding from the European Union's Horizon 2020 research and innovation program under the H2020-FETOPEN-2018-2020 grant agreement no. [899673] "Meta-fast" (T.P., I.S., G.S. and G.D.V.). This work reflects only the author view, and the European Commission is not responsible for any use that may be made of the information it contains. This work was also partially funded by the Deutsche Forschungsgemeinschaft (DFG, German Research Foundation) through the Collaborative Research Centre SFB 1375 "NOA" (T.P., G.S. and I.S.), the International Research Training Group (IRTG) 2675 "Meta-Active", project number 437527638 (T.P., I.S. and G.S.) and through the Emmy Noether Program, project number STA 1426/2-1 (I.S.).

## Author contributions

A.S., S.H.L., F.C. and S.K. performed the optical measurements. D.Y.C. and S.K. fabricated the meta-optical system. S.H.L., F.C. performed theoretical calculation. A.S. and I.S. wrote the first draft of the manuscript. I.S., T.P., G.S. and G.D.V. funding acquisition. I.S., G.S. and G.D.V. proposed the idea. I.S., T.P., G.S. and G.D.V. supervision and project administration. All authors analyzed the results and contributed to the writing and discussion of the manuscript.

## Funding

## Competing interests

The authors declare no competing interests.
