## [Peer Review File · Nature Communications]

REVIEWER COMMENTS

Reviewer #1 (Remarks to the Author):

In this work, the authors experimentally demonstrate a meta-optical system that enables complex control of the wavefront of light with pulse-duration limited dynamics. By combining the ultrafast polarization switching in a WSe₂ monolayer and a dielectric metasurface, the deflection of second harmonic waves with specific orbital angular momentum are realized on the femtosecond timescale. The presented results may pave the way for free space optical communications. The proposed idea is very novel and successfully verified. I would like to support the publication of this work in Nature Communications after revisions.

Suggestions and comments:

1. Although the results in Figure 2b and 2c have been studied in a previous work, it is better to briefly discuss the mechanism of polarization switching.
2. The fabrication process of the dielectric metasurface should be included in the methods section.
3. Why the calculated mode pattern in Figure 3c is also nonuniform.
4. What is the diffraction efficiency of the dielectric metasurface, it is better to discuss it in the main text.
5. In figure 4, if the authors have scanned a longer range, it is better to plot evolution of Gaussian-Vortex Gaussian-beam profiles.

Reviewer #2 (Remarks to the Author):

This manuscript uses polarization-controlled second harmonic generation in a TMD monolayer, which is then made to pass through a polarization-sensitive metasurface for wavefront shaping. The authors demonstrate ultrafast control using a femtosecond-level delay line of beam deflection, Gaussian-to-vortex beam switching, and topological charge switching.

This is a relatively interesting and likely significant manuscript that proposes ultrafast control of second harmonic generation as a potential way to modulate AOM. It is a succinct manuscript, with some flaws originating in its brevity. I have some critical questions to assess the relevance and interest of the manuscript properly.

1. Why are TMDs really advantageous for this application? The authors mention crystal symmetry and ultrathin nature for phase matching as advantages, but these are not unique properties for TMDs as nonlinear optical materials. Please elaborate on the advantages (and disadvantages) of TMDs for this specific nonlinear application. Why wouldn't one use a thin film of a conventional material instead? In principle, the TMD could even be replaced by a bulk SHG crystal.

2. The introduction seems to promise ultrafast modulation of OAM, but in the end, the changes in output are due to a tunable delay line, which does not provide fast switching/modulation. I found such a description to be unfair and possibly only geared to inflating the apparent relevance of the work. Indeed, the authors state in the abstract that they “demonstrate second harmonic beam deflection and structuring of orbital angular momentum on the femtosecond timescale”, but this statement is applicable to any second harmonic process with a similar delay line, where delay can be controlled very accurately “on the femtosecond timescale” while not providing effective modulation useful for high-speed communications. In summary, there is a faulty logical link between the motivation and what is finally shown. Please comment on this aspect and maybe suggest if there is any way out of it to actually provide the promised ultrafast wavefront shaping for the “dynamic generation of topological charges, as opposed to the simpler modulation of intensity or temporal phase.”

3. Related to the previous question, the conclusions contain the statement “More generally, the demonstrated speed-up in the synthesis of complex arbitrary wavefronts by more than 6 orders of magnitude”. It seems to compare different synthesis times/speeds: on the one hand, the modulation speed of SLMs and DMDs; on the other hand, the time delay used for the two polarizations. These are different quantities with different applications, so the comparison is not fair.

4. There is a paragraph about TMDs and their all-optical modulation of SHG: “However, despite its far-reaching potential, the mechanism has up to now only been employed for polarization and amplitude modulation.” Please provide references for the polarization and amplitude modulation demonstrated capabilities.

5. The caption of Figure 1 requires some additional text for the figure to be understandable as a stand-alone object.

6. Please clarify the meaning of the sentence “This is the case since the local phases for the two different polarizations have no simple relation as for the other considered phase profiles”.

7. Please spell out the fabrication methods in the Methods or Supplementary Information to guarantee accessibility to the recipe and reproducibility.

8. It is necessary to provide more information about the metasurface designs. Showing an image is not enough for reproducibility. Nowadays, sharing such complex design patterns should be compulsory if a simple description cannot be provided based on a few parameters. Both the design process and the final design obtained should be described.

9. In Figures 3c, f, j, m, there is a qualitative disagreement between theory and experiments for topological charge switching. Can the authors comment on the origin of that disagreement? Is it related to the method based on a cylindrical lens not being considered quantitatively?

10. The manuscript indicates that the pump-probe Fourier microscope is described in Methods, but in reality, it is found in the supplementary information figure S4, where no written description is given next to it.

Reviewer #3 (Remarks to the Author):

Sinelnik et al. report on the ultrafast optical switching of the OAM and other spatial characteristics of light pulses in a complex metaphotonic system, whereby an interferometrically controlled femtosecond pump generates polarization-controlled second harmonic (SH), which is in turn sent to a dielectric metasurface that can perform various spatial transformations on the SH beam. This is an elegant paper demonstrating a unique functionality not shown before, representing a significant advance in nanophotonics, materials science, and engineering. I recommend further consideration for Nature Communications when the following issues are resolved.

The primary concern I have about data presentation is that at first, by reviewing Fig.1 and the associated description and the text, I was under the impression that near zero delay, the system deterministically generates a ZZ-polarized SH. This was utterly surprising as the state must be phase-sensitive and depend on whether the incoming AC and ZZ-polarized pump beams are in-phase or out-of-phase. Then, it became more apparent that the authors are only concerned with the situation where the SH is linearly polarized, essentially throwing away most of the temporal data for the intermediate steps where the polarization is elliptical. This way of data presentation is quite confusing: it turns out that time in Fig.4 is essentially discrete and not a continuously tuned quantity (see Fig. S5 for the full trace). I strongly suggest updating the way data is presented as far as the temporal tunability is concerned and disclosing the selective nature of their experiment in the very beginning.

Also, the authors claim “switching between any two arbitrary wavefronts,” which is a very general statement not supported in the text, where only several specific examples are given. I suggest softening the claim.

Additionally, there's a typo in Fig 1 ('AS' instead of 'AC'). For better presentation, I suggest flipping the AC pulses sideways so that they're visually distinguished from the ZZ pulses. Fig.4 shows 'size' for the axis label, but it is not clear what is the exact nature of the object whose size is being reported. Probably, 'coordinate' would be more appropriate for the label.

**REPLY TO REVIEWERS' REPORTS AND A SUMMARY OF THE CHANGES MADE
IN THE MANUSCRIPT (NCOMMS-23-47460-T)**

Reviewer #1 (Remarks to the Author):

In this work, the authors experimentally demonstrate a meta-optical system that enables complex control of the wavefront of light with pulse-duration limited dynamics. By combining the ultrafast polarization switching in a WSe2 monolayer and a dielectric metasurface, the deflection of second harmonic waves with specific orbital angular momentum are realized on the femtosecond timescale. The presented results may pave the way for free space optical communications. The proposed idea is very novel and successfully verified. I would like to support the publication of this work in Nature Communications after revisions.

Our response: We thank the Referee for the nice summary of our work and for supporting its publication in Nature Communications pending revisions.

Suggestions and comments:

Reviewer #1: *Although the results in Figure 2b and 2c have been studied in a previous work, it is better to briefly discuss the mechanism of polarization switching.*

Our response: We have now added the following additional text to the revised manuscript:

Considering the D_{3h} symmetry of TMD monolayers, the emitted SH intensity in the basis of the AC/ZZ axes can be written as a function of the incident electric fields along these crystallographic directions as³¹:

$$\begin{pmatrix} I_{AC}(2\omega) \\ I_{ZZ}(2\omega) \end{pmatrix} \propto \begin{pmatrix} |E_{AC}^2 - E_{ZZ}^2|^2 \\ 2|E_{ZZ}E_{ZZ}|^2 \end{pmatrix} \quad (1)$$

On the basis of Eq. 1, the polarization of the incident fundamental pulses can be used to manipulate the polarization of the SH signal: if the time delay Δt between the two orthogonally polarized excitation pulses exceeds their pulse duration τ , the SH signal will be polarized along the AC direction ($I_{AC}(2\omega) \neq 0, I_{ZZ}(2\omega) = 0$). On the other hand, the maximum SH emission

along the ZZ direction is achieved for a delay $\Delta t = 0$, where the two fundamental pulses overlap in the time domain, leading to vanishing SH emission along the AC direction ($I_{AC}(2\omega) = 0, I_{ZZ}(2\omega) \neq 0$).

Reviewer #1: *The fabrication process of the dielectric metasurface should be included in the methods section.*

Our response: We thank the Reviewer for the comment and we agree that our fabrication process for high quality metasurfaces should be further highlighted in the manuscript. We have thus added the following text to the methods:

The proposed metasurfaces were fabricated through a meticulous process on a 1 mm-thick fused silica substrate. Before fabrication, the substrate was cleaned using a combination of acetone, isopropyl alcohol, and deionized water in an ultrasonic bath to enhance its adhesion to the a-Si:H film. Following this, a 475-nm-thick layer of a-Si:H was deposited onto the substrate using SiH₄ (25 sccm) gas with helium (475 sccm) dilution in a plasma-enhanced chemical vapor deposition system (Plasmalab 100 from Oxford). The substrate temperature, operating pressure, radio-frequency power at 13.56 MHz were 200 °C, 1900 mTorr, and 40 W, respectively. To define the desired features, a positive electron beam resist, ZEP520A from Zeon Chemicals, was spin-coated onto the substrate. In addition, Espacer (300Z from Showa Denko) was applied to prevent charging during the electron-beam exposure. The nanostructures were written into the resist by electron beam lithography (Raith150), followed by the development in ZED-N50. A 45-nm-thick aluminum film was deposited onto the substrate through electron-beam evaporation (Temescal BJD-2000) and then patterned by removing the resist with a solvent (ZDMAC from Zeon Co.). These aluminum patterns served as a robust etch mask during a-Si:H plasma etching. The etching process was carried out using a fluorine-based inductively coupled-plasma reactive ion etching system (Oxford Plasmalab System 100). The etching condition was optimized by controlling the mixture of CHF₃ and SF₆ gases, pressure and bias/induction powers which resulted in the formation of nanopillars with a high aspect ratio and vertical sidewalls. Subsequently, aluminum wet etching solution was used to remove any residual aluminum from the patterned nanopillars.

Reviewer #1: *Why the calculated mode pattern in Figure 3c is also nonuniform.*

Our response: We thank the Reviewer for the question. After carefully double-checking our calculation procedures, we acknowledge an error in the data processing, which resulted in the non-uniformity of Figure 3c. The error was induced from the different permutation of

dimensions of the multi-dimensional matrix when exporting results from Lumerical FDTD. The corrected figure is given below and it has been updated in the manuscript.

Figure 1R. Vortex intensity distribution after passing through a cylindrical lens.

In this context, we also highlight that, in order to further improve the agreement between the theoretical and experimental results in Figure 3c, f, j, m (see also comments of Referee #2), we have refined the theoretical calculations using a model that describes the cylindrical lens in a more accurate manner. In comparison to the approach used in the previous version of the manuscript, where the effect of the cylindrical lens was considered effectively as a 1D Fourier transform, the new model considers the propagation of the vortex beam through the cylindrical lens to the free space with diffraction theory. The computation was performed in scalar approximation, while the cylindrical lens is treated as a thin element. The focal length of the cylindrical lens, 50 mm, and the distance from the cylindrical lens to the detection plane, 200 mm, were taken from the experimental setting. In order to account for the orientation of the cylindrical lens in the experiment, the cylindrical lens axis is rotated by 60 degrees with respect to the propagation axis. The calculation results are provided above.

We have thus added the new version of Figure 3 to the main text, and the following paragraph to the supporting information:

Theoretical calculation of vortex beam propagation through a cylindrical lens

The calculation considers the propagation of the vortex beam through a cylindrical lens to free space with diffraction theory. The computation was performed within the scalar approximation, and the cylindrical lens is treated as a thin element with phase profile $\phi = \frac{2\pi}{\lambda} (f - \sqrt{f^2 + l^2})$. l is the coordinate along the axis of the cylindrical lens, i.e. $l = x \cos \theta + y \sin \theta$, where θ is the rotation angle of the cylindrical lens. The focal length of the cylindrical lens, 50 mm, and the distance from the cylindrical lens to the detection plane, 200 mm, were taken from the

experimental setting. In order to account for the orientation of the cylindrical lens in the experiment, the cylindrical lens axis is rotated by $\theta = -60$ degrees about the propagation z axis.

Figure 3. **Theoretical (left) and experimental (right) far-field distributions.** **a-c, g-j** LCP and **d-f, k-m** RCP light transmitted through the polarization-dependent metasurfaces in Fourier space. The first row shows results for beam deflection, the middle row for Gaussian-to-vortex beam switching, and the bottom row for topological charge switching. For c, f, j, m propagation of the vortex beam through a cylindrical lens was additionally considered.

Also, to properly refer to the new Supporting Information in the manuscript, we modified the following sentence in the main text:

“Thus, to evaluate the sign of the output beam, we employ an astigmatic transformation, through which the intensity pattern of the vortex beams acquires the form of dark stripes, whose orientation changes with the sign of l (Fig. 3 C,F).”

It now reads:

“Thus, to evaluate the sign of the output beam, we employ an astigmatic transformation, through which the intensity pattern of the vortex beams acquires the form of dark stripes, whose orientation changes with the sign of l (Fig. 3 C,F)^{37,38} (See Supporting Information for details of the calculation).”

[37] Denisenko V., et al. Determination of topological charges of polychromatic optical vortices. *Optics Express*, **17**, 23374-23379 (2009)

[38] Bekshaev A. Y., Soskin M. S., Vasnetsov M. V. Transformation of higher-order optical vortices upon focusing by an astigmatic lens. *Optics Communications*, **241(4-6)**, 237-247, (2004).

Reviewer #1: *What is the diffraction efficiency of the dielectric metasurface, it is better to discuss it in the main text.*

Our response: We thank the Reviewer for this useful suggestion. Indeed, the diffraction efficiency is a quantity of high interest for many readers. The blazed grating characteristics of the metasurfaces designed for beam deflection are ideally suited for determining the diffraction efficiency. To this end, we calculated the ratio of the SHG intensity channeled into the relevant first diffraction order (+/-1 for $\Delta t \sim 0$ / $\Delta t > 0$, respectively) and the total transmitted SHG intensity. To further add a quantitative discussion of the switching contrast for this case, we also calculated the ratio between the intensities in each of the first diffraction orders for $\Delta t \sim 0$ and $\Delta t > 0$. The results are summarized in the main text of the manuscript:

“To determine the diffraction efficiency, we calculated the ratio of the SHG intensity channeled into the relevant first diffraction order (+1 (-1) for $\Delta t \sim 0$ ($\Delta t > 0$), respectively) and the total transmitted SHG intensity, yielding 0.55 ± 0.05 (0.57 ± 0.05), correspondingly. Moreover, taking the ratio between the intensities in the +1st (-1st) diffraction orders for both delay settings allows us to quantify the switching contrast as 0.95 ± 0.05 .”

Reviewer #1: *In figure 4, if the authors have scanned a longer range, it is better to plot evolution of Gaussian-Vortex Gaussian-beam profiles.*

Our response: We thank the Referee for the suggestion. However, we have to point out that the presented graphs already include the maximum possible range for the delay line that we used in our experiments. In fact, to achieve the interferometric precision of our time-dependent measurements, which we show in Fig. 4, we used an adapted version of a commercial common-path birefringent interferometer (GEMINI, NIREOS). Due to the internal optics of the system, which are explained in more detail in the work of Preda *et al.* [1], we are experimentally limited to temporal delays ranging from -200 fs to +2 ps. Thus, we are not able to go to the condition $\Delta t < -\tau$, where the two fundamental pulse replicas are again well separated in the time-domain and subsequently emit a SH signal along the AC direction. Therefore, it is not

possible with the current experimental setup to show the full Gaussian-to-vortex-to-Gaussian evolution.

[1] Preda, F., et al. Linear and nonlinear spectroscopy by a common-path birefringent interferometer. *IEEE Journal of Selected Topics in Quantum Electronics*, **23(3)**, 88-96, (2016).

Reviewer #2: (Remarks to the Author):

This manuscript uses polarization-controlled second harmonic generation in a TMD monolayer, which is then made to pass through a polarization-sensitive metasurface for wavefront shaping. The authors demonstrate ultrafast control using a femtosecond-level delay line of beam deflection, Gaussian-to-vortex beam switching, and topological charge switching. This is a relatively interesting and likely significant manuscript that proposes ultrafast control of second harmonic generation as a potential way to modulate AOM. It is a succinct manuscript, with some flaws originating in its brevity. I have some critical questions to assess the relevance and interest of the manuscript properly.

Our response: We thank the Reviewer for the nice and concise summary of our work and for his/her constructive comments, that we address in detail in the following.

Reviewer #2: *Why are TMDs really advantageous for this application? The authors mention crystal symmetry and ultrathin nature for phase matching as advantages, but these are not unique properties for TMDs as nonlinear optical materials. Please elaborate on the advantages (and disadvantages) of TMDs for this specific nonlinear application. Why wouldn't one use a thin film of a conventional material instead? In principle, the TMD could even be replaced by a bulk SHG crystal.*

Our response: We thank the Reviewer for the comment, which allows us to further stress the relevance of our work. We agree that the mechanism of polarization switching, which is based only on symmetry considerations and thus on the structure of the nonlinear susceptibility tensor, is not limited to 2D-TMDs. This was very recently demonstrated, for instance, by Di Francescantonio et al. [1] with the routing of upconverted light by approx. 400 nm thick AlGaAs metasurfaces. In this manuscript, the authors acknowledge our approach as the first evidence of ultrafast polarization switching at atomic scales. Following this consideration, one could argue that any ultra-thin nonlinear material with the same symmetry (D_{3h}) could give analogous

results. For instance, one could replace monolayer TMDs with thin multilayer 3R-TMDs [2], which can provide higher efficiencies while still operating in the “phase-matching-free” limit. However, we stress that bulk crystals, as suggested by the Reviewer, could not serve for our purpose. This is because as soon as the phase-matching becomes relevant in defining the NLO efficiency, the SH intensity along one direction will be orders of magnitude more intense compared to that in the orthogonal direction. Thus, our approach for ultrafast SH polarization switching is impossible in bulk materials.

Furthermore, compared to thin films of III-V semiconductors or other conventional materials, 2D TMDs offer distinct additional perspectives and advantages based on their unique properties. First and foremost, the 2D crystals offer easy pathways for integration into functional photonic (nano)architectures without the need for expensive and complex growth and fabrication technology. Also, their optical and optoelectronic response is dominated by excitons even at room temperature, offering important additional opportunities, e.g. with respect to electrical tuning. Finally, 2D TMDs exhibit interesting spin-valley physics, which could be exploited for valleytronics and can potentially allow for direct selective interaction with light beams carrying orbital angular momentum.

[1] Di Francescantonio A. et al. All-optical free-space routing of upconverted light by metasurfaces via nonlinear interferometry. *Nature Nanotechnology*. 2023 1-8. <https://doi.org/10.1038/s41565-023-01549-2>

[2] Xu X., et al. Towards compact phase-matched and waveguided nonlinear optics in atomically layered semiconductors. *Nature Photonics*, **16(10)**, 698-706, (2022).

To clarify these points, we have modified the manuscript in the following way:

“A new avenue for ultrafast all-optical switching is offered by the unique nonlinear optical properties of monolayers of transition metal dichalcogenides (TMDs)³⁰.”

was replaced by

“A new avenue for ultrafast all-optical switching is offered by the intriguing nonlinear optical properties of monolayers of transition metal dichalcogenides (TMDs)³⁰.”

The following sentence (including references) was added at the end of the introduction:

“We highlight that our approach for ultrafast polarization switching³¹, which provides one building block of the final device presented here, only works in absence of phase matching constraints. Thus, while the atomically thin TMD could be replaced by ultra-thin films (e.g., few layer 3R-TMDs³²), our approach for ultrafast SH polarization switching is prohibited in bulk materials. Furthermore, compared to metasurfaces based on thin films of III-V semiconductors²⁹ or other conventional nonlinear materials, 2D materials offer distinct additional perspectives and advantages: ease of integration on-chip³³, fiber³⁴ and nanostructured photonic architectures³⁵, excitonic response and tunability at room temperature³⁶, selective interaction with light beams carrying spin³⁷ and/or orbital³⁸ angular momentum.

[29] Di Francescantonio A., et al. All-optical free-space routing of upconverted light by metasurfaces via nonlinear interferometry. *Nature Nanotechnology*. 1-8, (2023).

<https://doi.org/10.1038/s41565-023-01549-2>

[32] Xu X., et al. Towards compact phase-matched and waveguided nonlinear optics in atomically layered semiconductors. *Nature Photonics*, **16(10)**, 698-706, (2022).

[33] He J., et al. Low-loss integrated nanophotonic circuits with layered semiconductor materials. *Nano Letters*, **21(7)**, 2709-2718, (2021).

[34] Li Y., et al. Nonlinear co-generation of graphene plasmons for optoelectronic logic operations. *Nature Communications*, **13(1)**, 3138, (2022).

[35] Mupparapu R., Bucher T., Staude I. Integration of two-dimensional transition metal dichalcogenides with Mie-resonant dielectric nanostructures *Advances in Physics: X* **5**, 1734083 (2020).

[36] Ross J. S., et al. Electrical control of neutral and charged excitons in a monolayer semiconductor. *Nature communications*, **4(1)**, 1474, (2013).

[37] Herrmann P., et al. Nonlinear All-Optical Coherent Generation and Read-Out of Valleys in Atomically Thin Semiconductors. *Small*, 2301126 (2023).

[38] Ishii, S, Yokoshi N., Ishihara H. Optical selection rule of monolayer transition metal dichalcogenide by an optical vortex. In *Journal of Physics: Conference Series* **1220**, 1, 012056, (2019, May).”

Reviewer #2: *The introduction seems to promise ultrafast modulation of OAM, but in the end, the changes in output are due to a tunable delay line, which does not provide fast switching/modulation. I found such a description to be unfair and possibly only geared to inflating the apparent relevance of the work. Indeed, the authors state in the abstract that they “demonstrate second harmonic beam deflection and structuring of orbital angular momentum on the femtosecond timescale”, but this statement is applicable to any second harmonic*

process with a similar delay line, where delay can be controlled very accurately “on the femtosecond timescale” while not providing effective modulation useful for high-speed communications. In summary, there is a faulty logical link between the motivation and what is finally shown. Please comment on this aspect and maybe suggest if there is any way out of it to actually provide the promised ultrafast wavefront shaping for the “dynamic generation of topological charges, as opposed to the simpler modulation of intensity or temporal phase.”

Our response: We thank the Reviewer for the important comment and for this opportunity to further clarify our claims. The Reviewer correctly observes that in our experiment the changes in output are induced by a tunable delay line, which by itself is slow and would clearly not allow for high data rates. However, ultrafast pump-probe experiments employing delay lines are the gold standard for the investigation of ultrafast phenomena in any system [1]. Key is that they allow for resolving and investigating processes at the femtosecond time scale (and even below). In our case, this process is the change of a complex wavefront. A pump probe experiment is crucially needed, as there is no other measurement technique fast enough to resolve the wavefront change. In this regard, our measurements clearly demonstrate that our device architecture can sustain ultrafast wavefront shaping down to fs timescales. Thus, we beg to disagree with the Referee and we firmly believe that there is no “unfair description” of our work. Here, we could refer to a practically unlimited literature for ultrafast all-optical switching (see e.g. Ref 2 and reference therein), for instance: ultrafast routing of upconverted light [3], ultrafast broadband dichroism in plasmonic metasurfaces [4], lightwave and PHz electronics (see e.g. Ref. 5 and references therein), ultrafast chirality logic gates [6], ultrafast all-optical switching in graphene [7] and in tin oxide films [8]. All these examples are based on “slow” pump-probe experiments, but still they demonstrated the ultrafast capabilities of all-optical devices.

Next, we would like to address the technological perspective. In our vision, the delay line itself would not be part of a future communication system. Instead, it would be replaced by a setting, where two stabilized, orthogonally polarized laser beams arrive simultaneously at the meta-optical system, with one of the beams featuring an amplitude modulation at high speed. This could e.g. be a pulse-to-pulse modulation for a femtosecond laser with high repetition rate, or, perspective and pending an increase in SHG conversion efficiency, an implementation employing a continuous-wave laser. For example, microring lasers [9] could reach repetition rates beyond GHz once mode-locked. Our pump-probe experiments emulate such amplitude modulation by use of the delay line. The “on”-state here corresponds to both pulses arriving at the meta-optical system simultaneously, while the “off” state corresponds to only a single laser pulse arriving at a time. Most importantly, this approach demonstrates that the meta-

optical system can transform a (trivial/binary) amplitude encoded information into a complex-wavefront encoded information, and our device can sustain any modulation speed down to fs (THz-PHz bandwidth). As such, we are not claiming that we have implemented a communication system encoding complex wavefronts with high data rates. However, we believe that we can legitimately claim ultrafast all-optical switching of complex wavefronts and discuss the potential applications of this capability in free-space optical communications. To avoid any misunderstanding and to enhance the fairness of our comparisons (see also next point), we have included the essence of the above discussion and refined the wording in our manuscript as follows:

“Specifically, we perform spatially resolved pump-probe experiments to demonstrate second harmonic beam deflection, vortex on-off switching and topological charge switching at femtosecond time scales, thereby increasing the speed of OAM modulators by six orders of magnitude.”

now reads

“Specifically, we perform spatially resolved pump-probe experiments to demonstrate the possibility of realizing second harmonic beam deflection, vortex on-off switching and topological charge switching down to femtosecond time scales.”

While the sentence:

“Importantly, our approach allows for switching between any two arbitrary wavefronts with pulse-duration limited dynamics.”

now reads:

“Importantly, given the capability of polarization-selective metasurfaces to imprint also more complex spatial phase profiles³⁹, our approach has the potential to enable switching between any two nearly arbitrary wavefronts with pulse-duration limited dynamics.

[39] Arbabi A., et al. Dielectric metasurfaces for complete control of phase and polarization with subwavelength spatial resolution and high transmission. *Nature nanotechnology*, **10(11)**, 937-943, (2015).”

[1] Weiner, A. M. (2011). *Ultrafast optics*. John Wiley & Sons.

- [2] Chai Z., et al. Ultrafast all-optical switching. *Advanced Optical Materials*, **5(7)**, 1600665, (2017).
- [3] Di Francescantonio A. et al. All-optical free-space routing of upconverted light by metasurfaces via nonlinear interferometry *Nature Nanotechnology*. 1-8, (2023)
<https://doi.org/10.1038/s41565-023-01549-2>
- [4] Schirato A., et al. Transient optical symmetry breaking for ultrafast broadband dichroism in plasmonic metasurfaces. *Nature Photonics*, **14(12)**, 723-727, (2020).
- [5] Yang Y., et al. Light phase detection with on-chip petahertz electronic networks. *Nature communications*, **11(1)**, 3407, (2020).
- [6] Zhang Y., et al. Chirality logic gates. *Science Advances*, **8(49)**, eabq8246, (2022).
- [7] Ono M., et al. Ultrafast and energy-efficient all-optical switching with graphene-loaded deep-subwavelength plasmonic waveguides. *Nature Photonics*, **14(1)**, 37-43, (2020).
- [8] Jiang H., Broad-Band Ultrafast All-Optical Switching Based on Enhanced Nonlinear Absorption in Corrugated Indium Tin Oxide Films. *ACS nano*, **16(8)**, 12878-12888, (2022).
- [9] Liu Y., et al. A photonic integrated circuit–based erbium-doped amplifier. *Science*, **376(6599)**, 1309-1313, (2022).

Reviewer #2: *Related to the previous question, the conclusions contain the statement “More generally, the demonstrated speed-up in the synthesis of complex arbitrary wavefronts by more than 6 orders of magnitude”. It seems to compare different synthesis times/speeds: on the one hand, the modulation speed of SLMs and DMDs; on the other hand, the time delay used for the two polarizations. These are different quantities with different applications, so the comparison is not fair.*

Our response: Following also the response to our previous comment, we have now refined our discussion by adding the following paragraph:

“Specifically, we envision that in a setting, where two stabilized and orthogonally polarized laser beams arrive simultaneously at the meta-optical system, the latter can transform any amplitude information encoded in one of the beams into a complex-wavefront encoded information without any delayed response down to fs timescales. Furthermore, and beyond the advantage in terms of high operation speed, we highlight that OAM states of light, compared to amplitude modulation, are exceptionally good performers for the encoding of digital information, being highly resistant to eavesdropping and offering a level of security that does not depend on mathematical or quantum-mechanical encryption methods⁴⁶.”

[46] Gibson G., et al. Free-space information transfer using light beams carrying orbital angular momentum. *Optics express*, **12(22)**, 5448-5456, (2004).”

Reviewer #2: *There is a paragraph about TMDs and their all-optical modulation of SHG: “However, despite its far-reaching potential, the mechanism has up to now only been employed for polarization and amplitude modulation.” Please provide references for the polarization and amplitude modulation demonstrated capabilities.*

Our response: We thank the Referee for the comment. We have now added the relevant reference as suggested:

“However, despite its far-reaching potential, the mechanism has up to now only been employed for polarization and amplitude modulation [31].

[31] Klimmer S., et al. All-optical polarization and amplitude modulation of second-harmonic generation in atomically thin semiconductors. *Nature Photonics*, **15**, 837-842 (2021)”

Reviewer #2: *The caption of Figure 1 requires some additional text for the figure to be understandable as a stand-alone object.*

Our response: We thank the Referee and we agree that a brief explanation in the figure caption will further clarify our proposed scheme.

We added the following caption for Fig. 1 to the main text of the revised manuscript.

“Schematic representation of the operating principle of the cascaded TMD-metasurface structure for ultrafast wavefront shaping. In the first step, the polarization axes of two orthogonally linear polarized pulse replicas with the same pulse duration τ are aligned along the main crystal axes (AC and ZZ directions, respectively) of the WSe_2 monolayer. As a result of the D_{3h} symmetry of the crystal lattice and the associated contributions of the nonlinear susceptibility, depending on the time offset (Δt) of the two pulse replicas, the signal of the generated second harmonic (SH) in the WSe_2 monolayer is emitted either along the AC ($\Delta t > \tau$) or the ZZ direction ($\Delta t = 0$). In the second step of the cascaded structure, a quarter-wave plate (QWP), whose fast axis is oriented at a 45° angle with respect to both, AC and ZZ direction, leads to a left (right) handed circular polarization of the SH for an emission along AC (ZZ), depending on the temporal delay in the aforementioned SHG process. Finally, the designed silicon metasurface manipulates the SH wavefront depending on the helicity of the

incident radiation, which, in the displayed example, leads either to a Gaussian or a vortex beam shape.”

Reviewer #2: *Please clarify the meaning of the sentence “This is the case since the local phases for the two different polarizations have no simple relation as for the other considered phase profiles”.*

Our response: We have added the following clarification in the manuscript:

“Conveniently, the PB phase for a given unit cell design changes the sign but preserves the amplitude for orthogonal circular polarizations, such that a simple reversal of a phase gradient, as required for the metasurfaces implementing beam deflection and topological charge switching (see below for details), is by default implemented.”

Furthermore,

“This is the case since the local phases for the two different polarizations have no simple relation as for the other considered phase profiles”.

was replaced by

“This is the case since the local phases for the two different polarizations are not connected via a simple sign flip as for the other considered phase profiles”.

Reviewer #2: *Please spell out the fabrication methods in the Methods or Supplementary Information to guarantee accessibility to the recipe and reproducibility.*

Our response: As already addressed in the context of comment #2 of reviewer #1, we have added the following text to the methods:

“The proposed metasurfaces were fabricated through a meticulous process on a 1 mm-thick fused silica substrate. Before fabrication, the substrate was cleaned using a combination of acetone, isopropyl alcohol, and deionized water in an ultrasonic bath to enhance its adhesion to the a-Si:H film. Following this, a 475-nm-thick layer of a-Si:H was deposited onto the substrate using SiH₄ (25 sccm) gas with helium (475 sccm) dilution in a plasma-enhanced chemical vapor deposition system (Plasmalab 100 from Oxford). The substrate temperature, operating pressure, radio-frequency power at 13.56 MHz were 200 °C, 1900 mTorr, and 40 W,

respectively. To define the desired features, a positive electron beam resist, ZEP520A from Zeon Chemicals, was spin-coated onto the substrate. In addition, Espacer (300Z from Showa Denko) was applied to prevent charging during the electron-beam exposure. The nanostructures were written into the resist by electron beam lithography (Raith150), followed by the development in ZED-N50. A 45-nm-thick aluminum film was deposited onto the substrate through electron-beam evaporation (Temescal BJD-2000) and then patterned by removing the resist with a solvent (ZDMAC from Zeon Co.). These aluminum patterns served as a robust etch mask during a-Si:H plasma etching. The etching process was carried out using a fluorine-based inductively coupled-plasma reactive ion etching system (Oxford Plasmalab System 100). The etching condition was optimized by controlling the mixture of CHF_3 and SF_6 gases, pressure and bias/induction powers which resulted in the formation of nanopillars with a high aspect ratio and vertical sidewalls. Subsequently, aluminum wet etching solution was used to remove any residual aluminum from the patterned nanopillars.”

Reviewer #2: *It is necessary to provide more information about the metasurface designs. Showing an image is not enough for reproducibility. Nowadays, sharing such complex design patterns should be compulsory if a simple description cannot be provided based on a few parameters. Both the design process and the final design obtained should be described.*

Our response: We thank the Reviewer for raising concern on reproducibility of the metasurface design for others. A two-page description of the design process of the metasurfaces was already provided in the section “Design principle of metasurface” and “Selection of meta-atoms” in the Supporting Information. In response to the Reviewer’s suggestion, the layouts of the final design of the fabricated metasurfaces in gds format and the simulation models in stl format were provided as supporting material through Figshare during resubmission.

The following modification is implemented at the end of the section “Selection of meta-atoms” in the Supporting Information:

“The models of the three metasurfaes constructed with the nanoposts according to the corresponding phase profiles are shown in Fig. S3.”

Was replaced by

“For all three metasurfaces, the 3D models with the selected nanopost dimensions ($r_x = 60$ nm, $r_y = 185$ nm) are available for download (see Data availability statement).“

Moreover, the following was added to the paragraph explaining the fabrication method:

“The fabrication layouts of the three metasurfaces are available for download (see Data availability).”

Reviewer #2: *In Figures 3c, f, j, m, there is a qualitative disagreement between theory and experiments for topological charge switching. Can the authors comment on the origin of that disagreement? Is it related to the method based on a cylindrical lens not being considered quantitatively?*

Our response: We are grateful to the Reviewer for his/her comment and question. Indeed, we observed a notable discrepancy between theory and experiment in Figure 3c, f, j, m of our original manuscript regarding the elongation axis of the whole pattern and the intensity distribution of the two bright fringes, which a result of the simplicity of the original theoretical approach.

To improve the agreement between experimental and simulated data, we refined our model to describe the cylindrical lens in a more accurate manner. In comparison to the original minimalistic method, where the effect of the cylindrical lens was considered effectively as a 1D Fourier transform, the new model considers the propagation of the vortex beam through the cylindrical lens to free space using diffraction theory. The computation was performed in scalar approximation, while the cylindrical lens was treated as a thin element. The focal length of the cylindrical lens, 50 mm, and the distance from the cylindrical lens to the detection plane, 200 mm, were adopted from the experimental setting. In order to account for the orientation of the cylindrical lens in the experiment, the cylindrical lens axis is rotated by 60 degrees about the propagation axis in the analysis. The calculation results are provided below, showing greatly enhanced agreement with experimental data (see also new Fig. 3 further below for a direct comparison).

Figure 2R. Vortex intensity distribution after passing through a cylindrical lens.

Fig. 3 was updated accordingly, and the following section was added to the Supplementary Information:

“Theoretical calculation of vortex beam propagation through a cylindrical lens

The calculation considers the propagation of the vortex beam through a cylindrical lens to free space with diffraction theory. The computation was performed within the scalar approximation, and the cylindrical lens is treated as a thin element with phase profile $\phi = \frac{2\pi}{\lambda} (f - \sqrt{f^2 + l^2})$. l is the coordinate along the axis of the cylindrical lens, i.e. $l = x \cos \theta + y \sin \theta$, where θ is the rotation angle of the cylindrical lens. The focal length of the cylindrical lens, 50 mm, and the distance from the cylindrical lens to the detection plane, 200 mm, were taken from the experimental setting. In order to account for the orientation of the cylindrical lens in the experiment, the cylindrical lens axis is rotated by $\theta = -60$ degrees about the propagation z axis.’

Figure 3. **Theoretical (left) and experimental (right) far-field distributions.** a-c, g-j LCP and d-f, k-m RCP light transmitted through the polarization-dependent metasurfaces in Fourier space. The first row shows results for beam deflection, the middle row for Gaussian-to-vortex beam switching, and the bottom row for topological charge switching. For c, f, j, m propagation of the vortex beam through a cylindrical lens was additionally considered.”

Also, to properly refer to the new Supporting Information in the manuscript, we modified the following sentence in the main text:

“Thus, to evaluate the sign of the output beam, we employ an astigmatic transformation, through which the intensity pattern of the vortex beams acquires the form of dark stripes, whose orientation changes with the sign of l (Fig. 3 C,F).”

It now reads:

“Thus, to evaluate the sign of l of the output beam, we employ an astigmatic transformation, through which the intensity pattern of the vortex beams acquires the form of dark stripes, whose orientation changes with the sign of l (Fig. 3 c,f)^{44,45} (see Supporting Information for details of the calculation).

[44] Denisenko V., et al. Determination of topological charges of polychromatic optical vortices. *Optics express*, **17**, 23374-23379 (2009)

[45] Bekshaev A. Y., Soskin M. S., Vasnetsov M. V. Transformation of higher-order optical vortices upon focusing by an astigmatic lens. *Optics Communications*, **241(4-6)**, 237-247, (2004).”

Reviewer #2: *The manuscript indicates that the pump-probe Fourier microscope is described in Methods, but in reality, it is found in the supplementary information figure S4, where no written description is given next to it.*

Our response: We thank the Reviewer for this comment. We have corrected the reference where to find the detailed description and added a descriptive sentence to Fig. S4 in the SI. A detailed description of the setup can already be found in the main text of the manuscript (reproduced below for convenience).

“Figure S3 shows a sketch of the pump-probe setup used to observe the ultrafast change in the wavefront (see main text for a detailed description).”

“For an experimental demonstration of the various ultrafast wavefront shaping effects, we constructed a dedicated pump-probe setup featuring a Fourier microscope to observe the ultrafast change in the wavefront (see Fig. S3 for a sketch). As light source, we used the signal beam of an optical parametric oscillator (OPO) laser system (Mira OPO, APE, Germany)

pumped by a femtosecond Ti:Sa laser (Coherent Chameleon Ultra II). The output wavelength was set to 1500 nm, the pulse repetition rate was 80 MHz, the pulse duration was 200 fs, and the power was 10 mW. We used a commercial common-path birefringent interferometer (GEMINI, NIREOS) to create two beams with orthogonal polarization and to control the time delay between them with attosecond precision and subwavelength interferometric stability. The superimposed beams were focused onto the WSe₂ monolayer using a Mitutoyo NIR M Plan 10x NA=0.26 objective, and the SH signal was collected using an identical objective. A short pass filter was used to filter out the beam at the fundamental harmonic frequency. Next, to convert the linear polarization of the SH beam (AC or ZZ) into circular polarization (LCP or RCP, respectively), a quarter-wave plate was employed. It should be noted that the orientation of the quarter-wave plate was fixed during the experiments, the polarization was changed only by the ultrafast nonlinear polarization switching effect in the WSe₂ monolayer as a function of the pulse delay. The circularly polarized SH beam was focused onto the metasurface using a Mitutoyo NIR M Plan 10x NA=0.26 objective, the transmitted signal was collected with a 20x NA=0.4 objective from the same manufacturer. Finally, a system of lenses was used to image the back focal plane of the collection objective onto an sCMOS camera (prime 95B, Teledyne, USA). For the case of topological charge switching, one of the lenses was replaced by a cylindrical lens. This lens introduces a controlled astigmatism, leading to different intensity distributions on the camera, depending on the sign of l ."

Reviewer #3: (Remarks to the Author):

Sinelnik et al. report on the ultrafast optical switching of the OAM and other spatial characteristics of light pulses in a complex metaphotonic system, whereby an interferometrically controlled femtosecond pump generates polarization-controlled second harmonic (SH), which is in turn sent to a dielectric metasurface that can perform various spatial transformations on the SH beam. This is an elegant paper demonstrating a unique functionality not shown before, representing a significant advance in nanophotonics, materials science, and engineering. I recommend further consideration for Nature Communications when the following issues are resolved.

Our response: We thank the Reviewer for his/her positive assessment of our work.

Reviewer #3: *The primary concern I have about data presentation is that at first, by reviewing Fig.1 and the associated description and the text, I was under the impression that near zero delay, the system deterministically generates a ZZ-polarized SH. This was utterly surprising as the state must be phase-sensitive and depend on whether the incoming AC and ZZ-*

polarized pump beams are in-phase or out-of-phase. Then, it became more apparent that the authors are only concerned with the situation where the SH is linearly polarized, essentially throwing away most of the temporal data for the intermediate steps where the polarization is elliptical. This way of data presentation is quite confusing: it turns out that time in Fig.4 is essentially discrete and not a continuously tuned quantity (see Fig. S5 for the full trace). I strongly suggest updating the way data is presented as far as the temporal tunability is concerned and disclosing the selective nature of their experiment in the very beginning.

Our response: We thank the Reviewer for the suggestion. To improve the presentation and avoid possible confusion, we have moved the image showing the full temporal trace from the Supporting Information to the main manuscript, where it is included as Fig. 4 (a).

New Figure 4. Ultrafast wavefront shaping dynamics. (a) Cross section of the far-field intensity as a function of the delay time Δt for the case of Gaussian-to-vortex beam switching (full interferometric trace). The inset shows a fragment of the interferometric fringes obtained from the region marked by the white dashed lines. (b) Reduced interferometric trace for frames corresponding to a linear output polarization only. The inset shows the intensity in the center of the map (dashed white line) versus time. The red curve represents a guide to the eye obtained by smoothing the experimental data using a Savitzky–Golay filter.

Also, Fig. S4 was adjusted accordingly to avoid redundancy with the main paper. Specifically, figure part (A) was deleted and the caption changed to:

Fig. S4 Polarization properties of interferometer output. Fragment of the full interferometric trace (compare inset of Fig. 4 (a)) showing a close-up of the interference fringes. The bottom images show the spatial SH intensity distribution for the delay values

marked in the trace. Doughnut shapes are clearly observed for delays corresponding to linear output states of the interferometer.

Moreover, the text in the main manuscript was adjusted to reflect the figure changes:

“Note that for overlapping pulses, the polarization of the GEMINI output changes systematically between linear, elliptical and circular polarization as the delay time is changed by less than an optical cycle. Thus, from the full interferometric trace we extracted only those frames which correspond to a linear output polarization (Fig. 4). The full interferometric trace is included in the Supporting Materials (Fig. S5).”

was changed to

“Note that for overlapping pulses, the polarization of the GEMINI output changes systematically between linear, elliptical and circular polarization as the delay time is changed by less than an optical cycle. The full interferometric trace is shown in Fig. 4 (a), additional information on the polarization of the output provided in Fig. S4. For further discussion, from the full interferometric trace, we extracted only those frames which correspond to a linear output polarization (Fig. 4 (b)).”

Reviewer #3: *Also, the authors claim “switching between any two arbitrary wavefronts,” which is a very general statement not supported in the text, where only several specific examples are given. I suggest softening the claim.*

Our response: We thank the Reviewer for careful reading and reflect on our corresponding thoughts below.

We can control the two phase profiles for the two polarizations at the output plane arbitrarily, because we utilize the combination of the Pancharatnam-Berry phase and the propagation phase, which together provide two degrees of freedom. Nonetheless, it is true that we cannot control the phase and polarization in the far field, which are also required in the description of wavefront. We have refined and softened the claim as suggested (see also reviewer #2 point #2):

“Importantly, our approach allows for switching between any two arbitrary wavefronts with pulse-duration limited dynamics.”

now reads

Importantly, given the capability of polarization-selective metasurfaces to imprint also more complex spatial phase profiles³⁹, our approach has the potential to enable switching between any two wavefronts with nearly arbitrary intensity profiles and pulse-duration limited dynamics.

[39] Arbabi A., et al. Dielectric metasurfaces for complete control of phase and polarization with subwavelength spatial resolution and high transmission. *Nature nanotechnology*, **10**, 937-943 (2015)

Reviewer #3: *Additionally, there's a typo in Fig 1 ('AS' instead of 'AC'). For better presentation, I suggest flipping the AC pulses sideways so that they're visually distinguished from the ZZ pulses.*

Our response: Thank you for your comment. We have modified Fig. 1 as recommended.

Reviewer #3: *Fig.4 shows 'size' for the axis label, but it is not clear what is the exact nature of the object whose size is being reported. Probably, 'coordinate' would be more appropriate for the label.*

Our response: Thank you for your comment. We have modified Figure 4 as recommended.

REVIEWERS' COMMENTS

Reviewer #1 (Remarks to the Author):

The authors have successfully addressed my previous comments and suggestions.

Reviewer #2 (Remarks to the Author):

The authors have addressed my concerns, and I support the publication of the manuscript.

Reviewer #3 (Remarks to the Author):

In this reviewer's opinion, the authors have satisfactorily revised their manuscript, which I'm happy to recommend for publication.